# The Impact of Policy Modifiable Factors on Inequalities in Rates of Child Dental Caries in Australia

**DOI:** 10.3390/ijerph16111970

**Published:** 2019-06-03

**Authors:** Sharon Goldfeld, Kate Louise Francis, Monsurul Hoq, Loc Do, Elodie O’Connor, Fiona Mensah

**Affiliations:** 1Centre for Community Child Health, Murdoch Children’s Research Institute, Royal Children’s Hospital, Melbourne 3052, Australia; monsurul.hoq@mcri.edu.au (M.H.); Elodie.oconnor@mcri.edu.au (E.O.); 2Department of Paediatrics, University of Melbourne, Melbourne 3010, Australia; Fiona.mensah@mcri.edu.au; 3Clinical Epidemiology and Biostatistics Unit, Murdoch Children’s Research Institute, Royal Children’s Hospital, Melbourne 3052, Australia; kate.francis@mcri.edu.au; 4Australian Research Centre for Population Oral Health, Adelaide Dental School, The University of Adelaide, Adelaide 5005, Australia; loc.do@adelaide.edu.au

**Keywords:** child dental caries, child oral health, dental health services, fluoridation

## Abstract

*Background*: Poor oral health in childhood can lead to adverse impacts later in life. We aimed to estimate the prevalence and population distribution of childhood dental caries in Australia and investigate factors that might ameliorate inequalities. *Methods*: Data from the nationally representative birth cohort Longitudinal Study of Australian Children (N = 5107), using questions assessing: The experience of dental caries during each biennial follow-up period (2–3 years to 10–11 years), socioeconomic position (SEP), and policy modifiable oral health factors. *Results*: The odds of dental caries were higher for children with lowest vs. highest SEP (adjusted OR (adjOR) 1.92, 95% CI 1.49–2.46), and lower where water was fluoridated to recommended levels (adjOR 0.53, 95% CI 0.43–0.64). There was no evidence of an association between caries experience and either reported sugary diet or tooth brushing. When SEP and fluoridation were considered in conjunction, compared to the highest SEP group with water fluoridation children in the lowest SEP with fluoridation had adjOR 1.54 for caries, (95% CI 1.14–2.07), and children in the lowest SEP without fluoridation had adjOR 4.06 (95% CI 2.88–5.42). For patterns of service use: The highest SEP group reported a greater percentage of service use in the absence of caries. *Conclusions*: Dental caries appears prevalent and is socially distributed in Australia. Policy efforts should consider how to ensure that children with dental caries receive adequate prevention and early care with equitable uptake.

## 1. Introduction

Dental caries is a highly prevalent and preventable child health condition, affecting up to 60–90% of school-aged children in most industrialized countries [1,2,3,4]. Childhood dental caries can be caused by unhealthy lifestyles (i.e., poor diet, nutrition, and oral hygiene), limited availability and accessibility of oral health services, and poor living conditions [1,5]. Other physiologically influential factors include: Factors affecting tooth germ formation, presence of natural protective factors, such as saliva, and exposure to a low level of fluoride [6,7,8,9]. Poor oral health can negatively impact a child’s ability to eat, speak, sleep, and socialize, which may then have adverse impacts later in life [10,11]. 

The National Child Oral Health Study 2012–2014 has shown that the average number of untreated decayed tooth surfaces per child in the Australian child population was 1.3 (each tooth was divided into five surfaces). This was highest in five to six year olds (1.5) compared to 9–10 year olds (1.0) [12]. While 12-year-old children in Australia have less caries experience compared to the OECD average, it is higher than Germany, the United Kingdom, Sweden, and Canada [13,14,15]. There has been a steady rise in dental caries in Australian children attending school dental services over the last decade [13,16,17].

Dental caries in childhood represents a serious and potentially preventable public health problem, particularly in families with low socioeconomic position (SEP) [10,12,18,19] and is distributed disproportionately according to social determinants [20]. The prevalence of untreated dental caries has remained unchanged across the world over a 20 year period. However, the burden of untreated dental caries is not evenly distributed across the globe [3]. The Royal Australasian College of Physicians (RACP) released a position statement in 2012 highlighting that “paediatric oral health is one of the areas of greatest health inequity in Australia and New Zealand” [10]. Australian research has shown that low SEP is associated with higher odds of parent-reported caries, infrequent tooth brushing, and non-use of dental services [21]. International research has also shown this association between SEP and dental caries [22,23,24].

The availability of dental services in Australia is complex. Medicare, Australia’s publicly funded health system, has, for the most part, restricted dental health coverage. While specialist medical and allied health services are funded in part by Medicare, the majority of dental reviews are self-funded or paid for by a private health fund, often with remaining out of pocket expenses [25]. Individuals spent 19.6% of their health expenditure on dental services in 2016–2017 [25]. In 2014, the Child Dental Benefits Schedule (CDBS) was introduced, which allows children aged between 2 and 17 years from low and middle-income families to benefit from basic dental services capped at $1000 over two years [26]. However, in the first year of implementation, only 29% of eligible children accessed the program [27]. Public dental practices have long waiting lists [28]. In 2013, there were approximately 56 dentists per 100,000 people in Australia. Around 10% (n = 1403) of employed dentists were specialists, and only 94 of these were paediatric dentists [13]. Ambulatory care sensitive conditions (ACSCs) are conditions for which hospitalization is thought to be avoidable with early disease management in settings such as primary care: In Victoria, dental ACSCs have the highest rate of all ACSC hospital admissions for children, and more than $9 million is spent on avoidable dental ACSCs for children each year [29]. These are admissions that would have been avoided with good community primary (dental) care.

Fluoridation of the water supply appears to be an effective approach through which the social gradient observed for dental caries can be lessened, although studies are often limited to observational and ecological approaches [30,31,32]. The United States Centers for Disease Control and Prevention lists water fluoridation as one of the ten great public health achievements of the 20th century [33]. In Australia, inequalities may be exacerbated by variable access to dental services and fluoridated water across rural and remote Australia compared to major cities [34,35,36]. Other population-wide measures such as oral health promotion to decrease sugary food exposure, and increased use of dental services and tooth brushing, show promise and have had substantial policy focus over time [37]. However, few studies have shown the cumulative benefit of these public health measures [38]. 

More recent research has implicated the transmission of bacteria as a risk factor with adequate antenatal treatment of dental caries in the mother being a potentially effective approach to preventing decay in children, although again intervention studies have been limited [39]. Finally, more targeted approaches to dental caries prevention such as fluoride varnish show distinct promise with randomized controlled trials showing on average a 27% reduction in caries experience [40].

Of some interest from a policy perspective is the consideration of the cumulative benefit of modifiable oral health factors, a number of which are amenable to clinical and public health interventions. Few studies have examined multiple exposures to these potentially modifiable factors [5,37]. An important question is whether these approaches show substantial cumulative benefit and in particular whether they ameliorate or exacerbate the social gradient. Using longitudinal data, the current study aimed to determine the prevalence and population distribution of dental caries over time for children up to the age of 10–11 years in a representative Australian population, specifically assessing the association with policy modifiable oral health factors that might ameliorate inequalities in oral health. 

## 2. Materials and Methods 

### 2.1. Data Source

The Longitudinal Study of Australian Children (LSAC) is a nationally representative sample of two cohorts of Australian children—the birth cohort (B-cohort) of 5107 infants, and the kindergarten cohort (K-cohort) of 4983 four-year-olds—each of which commenced in May 2004 [41]. The LSAC design and sampling methodology are documented elsewhere [41,42]. In short, a complex survey design, sampling by postcode, was used to select a sample that was broadly representative of all Australian children except those living in remote areas [42].

Data were collected on multiple aspects of children’s oral health and diet as well as family characteristics, and multiple information sources were utilized, such as parent interview and self-report questionnaires. Parent-report data in this study were drawn from responses by “the parent who knows the child best”, in most cases, this was the child’s biological mother. 

This paper draws on data from the B-cohort (51.2% male), collected when children were aged zero to one (Wave 1, n = 5107), two to three (Wave 2, n = 4606), four to five (Wave 3, n = 4386), six to seven (Wave 4, n = 4242), eight to nine (Wave 5, n = 4085), and 10–11 years (Wave 6, n = 3764, see Figure 1). The retention rate (73.7%) between Wave 1 and 6 compares favourably with the retention rates achieved by other comparable overseas studies [43]. The current analysis is restricted to families who participated in every Wave (n = 3441). Survey weights were applied in order for the findings to maintain the representation of the target sample, who were selected to be representative of the Australian population of families. Figure 1 illustrates the high proportion of this cohort providing oral health data at each Wave. The LSAC methodology was approved by the Australian Institute of Family Studies Human Research Ethics Review Board.

### 2.2. Measures

#### 2.2.1. Dental Caries Experience

Parents reported on the child’s oral health (two to three years to 10–11 years), using the same item at each time-point: “Has the study child <insert time period> had any of the following problems with his/her teeth?” (1) cavities or dental decay, (2) teeth pulled because of dental decay, and (3) tooth or teeth filled because of dental decay. The proportion of children having at least one event of dental caries experience for a given time period was identified, i.e., those who responded “yes” to any item for that period. From here on, this outcome variable is called ‘proportion with caries’. Previous research in children has indicated that parent-reported single-item indicators of their children’s oral health have satisfactory construct validity [21,44,45,46]. Missing teeth due to “accident causing breakage or loss of teeth” or due to children’s teeth naturally falling out were excluded. 

#### 2.2.2. Socioeconomic Position (SEP)

A composite measure of SEP was derived at each Wave, from two to three years to 10–11 years, from each parent’s self-reported annual income, highest education, and occupation level [47]. The resulting continuous score was categorized into quintiles from low SEP to high SEP at each Wave.

#### 2.2.3. Policy Modifiable Oral Health Factors

*Water fluoride level (classified at zero to one year (Wave 1)).* Water fluoride level was classified according to the postcodes of the children in infancy (age zero to one year), matched with a postcode-based database of water fluoride levels, maintained at the Australian Research Centre for Population Oral Health (ARCPOH). Year of commencement of water fluoridation at each postcode was also available in the database enabling accurate classification according to the survey year of the LSAC cohort participants. Levels were provided in three groups (<0.3 mg/L, 0.3–0.5 mg/L and 0.6–1.1 mg/L) and subsequently categorized into two groups which will be referred to as non-fluoridated which are areas with levels not meeting the guidelines (0 to 0.5 mg/L) and fluoridated which are areas with levels meeting the guidelines (0.6 mg/L–1.1 mg/L). The cut point is the level recommended as the minimum provision for preventing caries (0.6 mg/L) [34]. 

*Remoteness (classified all survey Waves from two to three years).* Using the accessibility/remoteness index of Australia [48], children’s local areas were categorized as a major city, inner regional, outer regional, remote, or very remote area, to provide an indicator of geographic remoteness. 

*Sugary diet (classified all survey Waves from two to three years).* Parents reported on the frequency of consumption (in the last 24 hours) of various food and beverage items: (1) drink fruit juice, (2) eat biscuits, doughnuts, cake, pie or chocolate, and (3) drink soft drink or cordial, not diet. A value of zero (if they reported the item was not consumed at all), one (if they reported having the item once), or two (if the item was consumed twice or more) was assigned for each item. A sugary diet score ranging from zero to six was generated, with higher scores indicating a higher frequency of reported consumption of sweet food and/or beverages. 

*Tooth brushing age (classified at two to three years (Wave 2)).* Parents were asked: “How old was the study child when you started to clean his/her teeth?” 

*Tooth brushing frequency (classified all survey Waves from two to three years)* How frequently teeth were brushed was also recorded in the following categories: More than twice a day, twice a day, once a day, less than once/ not at all. For the analysis the categories were dichotomized based on the recommendation of brushing teeth twice a day, resulting in the variable of teeth brushing frequency: Twice a day or more, or less than twice a day. At ages four to five, there were largely incomplete data as parents were not asked to report how often the child cleaned their own teeth rather than parents cleaning the child’s teeth. Data for this Wave were replaced with the data from age six to seven years as a proxy to enable this variable to be included in all years in the analysis. There were no details available regarding the use of toothpaste. In Wave 6 (10–11 years) the child was asked directly about tooth brushing frequency via the question: “Yesterday, how often did you brush your teeth?” 

*Use of dental services (classified for survey years from six to seven years (Waves 3,4,5)).* Parents were asked: “In the last 12 months, have you used dental services for the study child?” However, dental visits were reported for the preceding 12 months and caries over the preceding two years, and so we were unable to differentiate preventive use from dental visits in response to decay. Although this meant that the service use data could not be used meaningfully as a predictor of caries in the analyses, social patterns in service use in relation to caries experience were examined to investigate patterns, and inequities, in service provision.

#### 2.2.4. Demographic Variables

Children’s age was recorded with each Wave of the survey, and gender reported by parents at survey Wave 1 was used in the analyses.

### 2.3. Statistical Analysis

All analyses were conducted using Stata version 15.1 (StataCorp LP., College Station, TX, USA). SEP and demographic characteristics, for children aged between two and three years and 10–11 years, and water fluoridation in infancy were summarised. The proportions of children with dental caries were estimated according to SEP quintile at each Wave. Survey weights were applied to account for the unequal probability of families initially participating in the study and subsequent sample attrition, and all estimates were corrected for the study design which sampled families according to postcode of residence. 

Multilevel mixed-effects logistic regression was used to investigate the association between the proportion with caries and each of SEP, water fluoride level, remoteness, age started cleaning teeth, sugary diet, and age, estimating odds ratios with 95% confidence intervals. All Waves of data were pooled, and the repeated measures of caries at each Wave were examined for each child. The postcodes and child unique identification numbers were used to identify the area levels and the child levels within the multilevel model to account for the sampling design and within child repeated measures. Next, the associations between SEP and dental caries according to water fluoride level were examined to investigate whether water fluoridation may ameliorate the socioeconomic gradients in proportion with caries which may be expected if children are not exposed to water fluoridation. Finally, dental service use patterns were examined in relation to children’s experience of caries and SEP. 

## 3. Results

At two to three years, the average age of the children was 34 months (Table 1), and 2.8% of children reported events of caries in the previous two years. This increased to 26.5% by the time children were 10–11 years. Around 28% of the children lived in non-fluoridated areas as infants (age zero to one). Daily frequency of tooth brushing was completed by approximately two-thirds of the sample from Wave 3 onwards. Around 10% of parents reported no consumption of sugary items in their child’s diet in the last 24 hours across each Wave of data collection (range: 9.1% (Wave 3) to 11.7% (Wave 6)). The mean sugary diet score of 2.1 to 2.2 across Waves indicated that children were on average consuming sugary food/ drinks twice or more per day. 

### 3.1. Dental Caries over Time

The parent-reported oral health data that were available for the analytic cohort of 3441 children from two to three years to 10–11 years are presented in Figure 1. Variation in the proportion of children experiencing dental caries according to SEP is illustrated in Figure 2. At two to three years, the proportion of children with caries was 2.8% and varied a little by SEP. By four to five years, the proportion of children with caries ranged from 7.4% (95% CI 5.7–9.3) for children from families with the highest SEP to 21.2% (95% CI 17.2–25.2) for children from families with the lowest SEP. At age six to seven years the proportion of children with caries in the highest SEP group was 18.2% (95% CI 15.4–21.0) compared to 30.4% (95% CI 25.6–35.2) in the lowest SEP group. At age eight to nine years when the highest proportion of children with caries was reported, that proportion in the highest SEP group was 24.4% (95% CI 21.4–27.3) compared to 33.6% (95% CI 29.3–37.9) in the lowest SEP group. The proportion of children with caries then started to drop for children at the age of 10–11 (cohort wide prevalence of 26.5% compared to 30.7% in the previous survey) for all quintiles of SEP, perhaps reflecting the natural exfoliation of deciduous teeth and the slower rate of caries in permanent teeth. 

### 3.2. Factors Associated with Dental Caries Experience

Table 2 describes multilevel mixed-effects logistic regression where each of SEP, water fluoride level, remoteness, age started cleaning teeth, sugary diet, and age are considered together as predictors of dental caries at each age, estimating adjusted odds ratios (adjOR) with 95% confidence intervals (CI). In comparison to children from families with the highest SEP, the odds of having caries was higher for children from the lowest SEP group (adjusted OR (adjOR) 1.92, 95% CI 1.49–2.46, Table 2). The odds of having caries were lower for children who lived in fluoridated areas as infants, compared to non-fluoridated areas (adjOR 0.53, 95% CI 0.43–0.64). Children living in very remote areas had higher odds of having caries compared to children living in major cities (adjOR 2.41, 95% CI 1.01–5.75). There was no evidence of an association between the proportion with caries and tooth brushing frequency, age children started cleaning their teeth or sugary diet.

Table 3 and Figure 3 illustrate the adjusted odds ratios for caries associated with the conjunction of the child’s SEP quintile and whether they lived in an area where the water was fluoridated as an infant. The combination of the highest SEP quintile and water fluoride at the guideline level is chosen as the reference group for comparison because these children are likely to be least at risk of caries for both of these reasons. These results indicate that children with lower SEP have consistently higher odds of caries than those with the highest SEP, however the SEP-gradient in caries is steeper for children who lived in areas where the water is non-fluoridated. Children in the lowest SEP quintile where water is non-fluoridated have almost four-fold odds of caries compared to children in the highest SEP quintile where water is fluoridated (adjOR 4.06, 95% CI 2.88–5.47). Amongst areas where water is fluoridated the disparity in caries between the children in the lowest and highest SEP quintiles is evident but is more tempered (adjOR 1.54, 95% CI 1.14–2.07). 

### 3.3. Service Use

Figure 4 illustrates the use of dental services with regard to the presence and absence of dental caries, reflecting the previously described four groups: Those who used dental services who reported no caries experience. Those who used dental services and had caries experience, those who did not use dental services and who reported no caries experience, and those who did not use dental services and had caries experience. Figure 4 shows that for children with higher SEP there were greater rates of dental service use in the absence of caries experience. Amongst families with the highest SEP, service use where no caries had been experienced was reported for 50.2% of children, indicating a dental visit for a check-up as a recommended preventive measure. This compares to only 23.8% of children from families with the lowest SEP. Children from the lowest SEP groups were also more likely to report that they had experienced caries in the previous two years but not used dental services over the last 12 months evidencing the gap in dental care for these children. 

## 4. Discussion

This study describes the prevalence and population distribution of childhood dental caries for children from two to three years up to the age of 10–11 years in a birth cohort sample representative of the Australian population, specifically assessing the association with policy modifiable oral health factors that might ameliorate inequalities in oral health. Rates of caries experience (decayed, missing, filled teeth) increased from 2.8% at two to three years to 26.5% at 10–11 years. The odds of having caries were highest for children living in families with the lowest SEP, in non-fluoridated areas, and in very remote areas. Although social disparities in caries were evident within fluoridated areas, the gradient was substantially ameliorated compared to the gradient observed in non-fluoridated areas. Further, for children from families with higher SEP, there were greater rates of dental service use in the absence of caries, the recommended preventive measure. Children from the lowest SEP group were more likely to report no use of dental services in the presence of caries. 

The findings from the current study reinforce the consistent findings from previous studies in Australia and internationally, which have shown the association between SEP and dental caries (e.g., [21,22,23,24]). However, the current study extends these findings by examining: 1) Exposure to fluoride at guideline levels followed up in children from zero to one years old (Wave 1) to 10–11 years old (Wave 6), 2) other SEP-related risk factors, such as rurality, 3) the interaction between adequate fluoride exposure and the amelioration of the social gradient, and 4) the potential inverse care law at play for children from poorer backgrounds in regards to use of and access to dental services, both for intervention and prevention of decay.

### 4.1. Fluoridation of the Water Supply

During the waves of the cohort, over 70% of the Australian population received fluoridated water ranging from 0.6 to 1.1 mg/L. Australia mostly has very low levels of naturally occurring fluoride. There are very few areas with natural fluoridation, mostly in remote or very remote areas (central Australia, remote areas of Queensland). Natural fluoridation levels range from 0.5 to 1.7 mg/L. In regions where levels of naturally occurring fluoride fall within Australia’s recommended range, water suppliers do not add fluoride to the water supply [34]. The study sample did not include children from those areas. 

The disparity in dental caries according to water fluoridation meeting guideline levels was more evident amongst children living in families with the lowest SEP, reflecting the potential for a universal intervention which may be effective in addressing inequity in oral health. Previous studies have examined the relationship between SEP and dental caries, and discussed the potential of water fluoridation to reduce the social gradient [30,31,32]. These have included a number of international and Australian studies suggesting the impact of fluoridation in ameliorating the social gradient and, in particular, differentially impacting more disadvantaged children (although details on actual levels of fluoride in the water supply are not always included). The 2016 Australian National Health and Medical Research Council Information Paper on the effects of water fluoridation on dental and other human health outcomes [34] evaluated contemporary evidence from observational studies with GRADE (grades of recommendation, assessment, development and evaluation) assessed quality. It concluded that there was consistent evidence to support the beneficial impact of water fluoridation for all socio-economic groups, however, the evidence to support amelioration of the social gradient through differential benefit was inconsistent and studies of variable quality [34]. In Australia Armfield [31] found that the most economically disadvantaged five to six year old children had 48% less decay if they lived in an optimally fluoridated area, compared to non-fluoridated areas. For Indigenous children, caries experience was 51% lower. A recent study using data of a population-representative sample of over 21,000 children from the National Child Oral Health Study 2012–2014 reported a significant differential benefit of exposure to fluoridated water for low-income and Aboriginal children [49]. This study investigated race- and income-related inequalities in dental caries experience of Australian children stratified by fluoridation status. Caries experiences were compared between fluoridated and non-fluoridated areas within stratifications by Indigenous status and quartiles of equivalised household income. Absolute and relative socioeconomic inequalities were estimated using the absolute concentration index (ACI), slope index of inequality (SII), relative concentration index (RCI), and relative index of inequality (RII). The study reported far greater levels of child caries experience among Indigenous and lower-income households than their counterparts. It also reported that Indigenous status- and income-related inequalities for the deciduous and permanent dentitions were consistently lower in fluoridated areas. A study from New Zealand, using administrative record data, found that water fluoridation reduced but not eliminated the gap in dental caries experience between Maori and non-Maori children [50].

Complicating this relationship is the challenge of water fluoridation in small rural and remote communities in Australia, where the impact of fluoridation is greater for people living outside capital cities taking all other factors, such as socio-economic status and dental visits, into account [51,52]. In the current study, the odds of having dental caries were higher for children living in outer regional and very remote areas, compared to children living in major cities. This finding is consistent with previous research [36]. This may be due to the lack of availability of dental services, and greater distances involved [35]. For example, it has been shown that most children living in rural and remote areas of Western Australia do not access a dental service before five years [51]. Perhaps a more effective use of the local workforce is needed. This could be the case for Queensland, Australia, where the proportion of outpatient oral services delivered to children up to the age of 14 years living in rural communities was less than the whole of Queensland [53]. 

### 4.2. Oral Health Promotion Factors

The findings of a lack of association between dental caries and the usual targets of oral health promotion, such as tooth brushing frequency, the age children started cleaning their teeth, and sugary diet aligns variably with the literature. Previous evidence has shown some benefits of tooth brushing frequency, lower intake of sweets and soft drinks, and the age children start cleaning their teeth in reducing caries in children [54,55,56]. In particular, one study showed a dose-response relationship between the prevalence of caries and brushing frequency in children [55]. Other studies which have looked at the overall effect of school-based tooth-bushing programs on caries experience have shown mixed results [57,58]. Associations between dental caries and other oral health promotion factors were not evident in our study but could be due to the use of self-reported data which has its limitations (see Section 4.4). 

Caution needs to be taken when interpreting the evidence as previous research has noted the potential for social desirability bias in reporting [59,60,61]. In particular, one study showed there was discordance in caregiver reporting child tooth brushing frequency which was likely due to social desirability bias [61]. Another showed low agreement between observed tooth brushing in children in comparison with that reported by their mothers [60]. 

### 4.3. Dental Services

There was evidence of a potential inverse care law at play [62,63], that is, children with potentially greater need having less access to quality services. We found children from the lowest SEP families being least likely to access dental services, both for intervention and prevention of decay. This is in contrast to higher SEP groups where there was relatively high access to dental service regardless of dental caries. This is supported by the National Child Oral Health Study which showed 21% of children had an irregular pattern of dentist visiting which was associated with lower SES, particularly Indigenous status, parental education, and low household income [12]. Similarly a study looking at predictors of dental visits among primary school children in rural Australian community found that private health insurance (as a proxy for higher income) and age were significantly associated with both 6-month and 12-month dental visits [52].

These data are replicated for adults in Australia with lower income groups having higher rates of unfavourable dental visiting patterns (visiting less than once every two years) than higher income groups, ranging from 35.1% for the $30,000 or less household income group to 12.4% for the $140,000 and over income group [13]. In a random sample of adults aged 30–61 years living in Australia dental insurance and income were positively associated with regular dental visiting. Individuals in the lowest income tertile had a lower prevalence of regular visiting than those in the highest income group [64]. This is not surprising given the restriction of dental health coverage in Medicare (Australia’s publicly funded health system). In comparison, UK data from the Children’s Dental Health Survey 2003 showed there were no socioeconomic differences in the use of preventative dental services (fissure sealants or any treatment to prevent caries) [65]. In the UK, free dental care through the National Health Service (UK’s government-funded medical and health care services) is given to children aged under 18 years [66].

### 4.4. Strengths and Limitations

This study capitalized on a large, nationally representative community sample of Australian children. This was a major strength of the study, as it allowed us to explore the prevalence and population distribution of dental caries for children from two to three years up to the age of 10–11 years, and to achieve greater generalizability of the current findings. In addition, we were able to use a mixture of parent-report and direct assessment (i.e., water fluoridation level according to postcode), adding to the robustness and ecological validity of the findings, that is, they reflect the real-world context of the findings. Nevertheless, in utilizing this valuable and rich existing data, we were limited by the available measures.

Research has detailed the satisfactory validity of parent-reported single-item indicators of their children’s oral health [44,67] and their robustness across socioeconomic circumstances [45]. However, as disease levels increase, the accuracy of using parental report could be better [67] and there is also a tendency towards under-reporting very young children’s oral health problems [44]. As such, it is likely there is an under-estimation of caries experience for the earlier Waves of this study, given the reliance on parent-reported oral health outcomes for their children. Future longitudinal research could endeavour to validate parent-reported oral health outcomes for their children against dental clinical examinations or dental records [21]. Additionally, children from very remote areas and Aboriginal and Torres Strait Islander children are not well represented in LSAC and care should be taken in extrapolating from these findings [68]. It is important that future studies should be as inclusive as is feasible and to ensure routine data systems extend to rural areas. Information on fluoridated toothpaste was not available. However, as fluoridated toothpaste comprises over 95% of the market in Australia, its inclusion is not expected to influence the study findings. Finally, water fluoride level was based on the residential postcode when the child was an infant (age zero to one year), and families may have moved into or out of areas where the water fluoride level met the guideline levels.

## 5. Conclusions

Dental caries is prevalent and socially distributed in Australia. Fluoridation of the water supply to guideline levels seems to ameliorate (although not eliminate) the social gradient and is an important and policy sensitive lever for addressing inequalities in caries. The service inequities appear stark yet unsurprising given the differential access through the current Australian dental care policies. Although the evidence to support dental services for preventive care is slim, the need to ensure that children with dental caries receive adequate and early care seems an important population preventive position. Our findings suggest current inequalities in dental caries may well be preventable. Policy efforts to ameliorate oral health inequalities should consider multiple strategies to ensure there is adequate distribution and equitable uptake of effective interventions.

## Figures and Tables

**Figure 1 ijerph-16-01970-f001:**
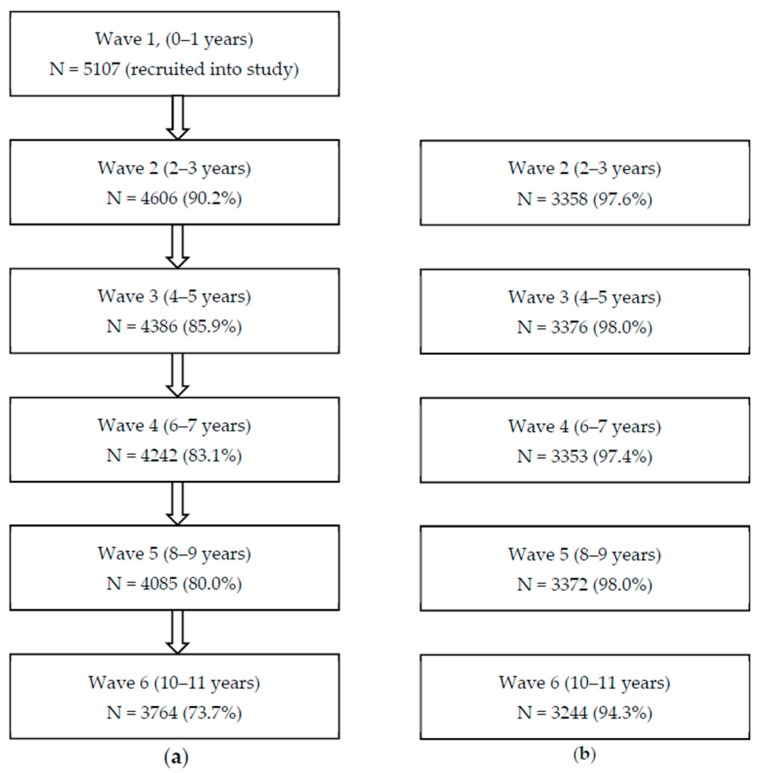
**(a**) Flowchart of cohort attrition for LSAC, (**b**) the number of participants who completed all the oral health questions and were the sample for this study, and percentage of the analytic cohort who had participated every year (n = 3441).

**Figure 2 ijerph-16-01970-f002:**
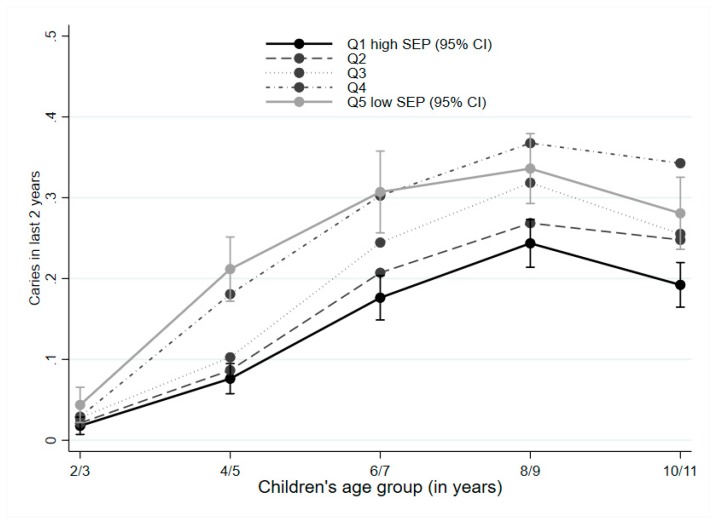
Proportion of children with caries in last two years, according to socioeconomic position.

**Figure 3 ijerph-16-01970-f003:**
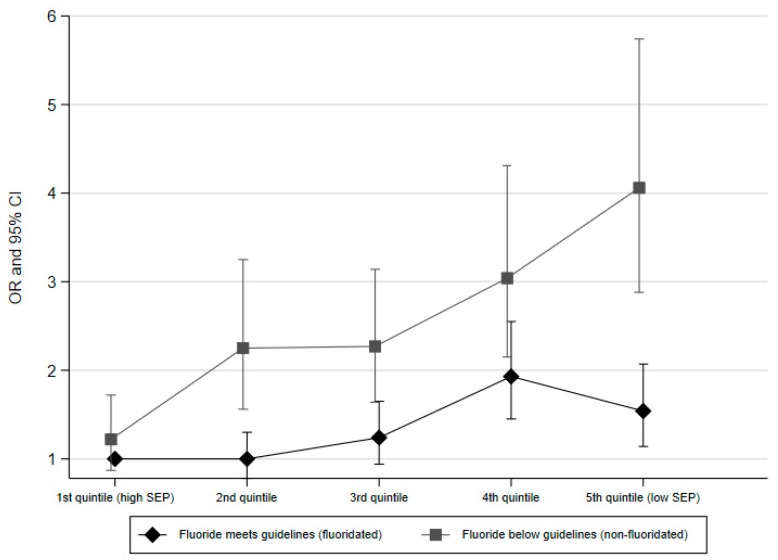
The odds ratio and 95% confidence intervals for each SEP quintile for those who live in fluoridated or non-fluoridated areas.

**Figure 4 ijerph-16-01970-f004:**
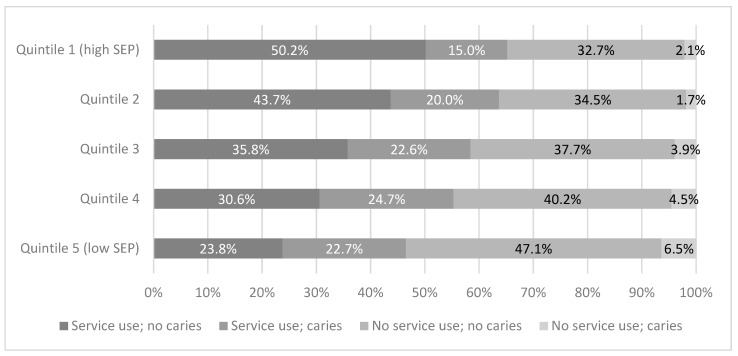
Service use in the previous year with relation to caries experience reported in the previous two years for each SEP quintile at age eight to nine years.

**Table 1 ijerph-16-01970-t001:** Characteristics of children and parent-reported oral health data (weighted to representative Australian sample).

	Wave 2	Wave 3	Wave 4	Wave 5	Wave 6
**Age in months (mean (SD))**	33.8 (4.0)	57.6 (3.3)	81.9 (4.4)	107.2 (4.5)	131.1 (5.2)
**Gender (female) (%) ^1^**	48.4	48.4	48.4	48.4	48.4
**Caries experience**					
No caries experience in last two years (%)	97.2	86.8	75.3	69.3	73.5
Had caries experience in last two years (%)	2.8	13.2	24.7	30.7	26.5
**SEP ^2^**					
Quintile 1 (high SEP) to Quintile 5 (low SEP) (%)	Each 20.0	20.0	20.0	20.0	20.0
**Remoteness**					
Major cities (%)	68.5	68.0	66.8	66.1	65.7
Inner regional (%)	19.0	20.3	21.1	22.2	22.5
Outer regional (%)	11.2	10.6	10.9	10.5	10.6
Remote (%)	0.8	0.7	0.8	0.9	0.9
Very remote (%)	0.4	0.4	0.3	0.4	0.4
**Water fluoride level: Residential area in infancy ^1^**					
≤ 0.6 mg/L (non-fluoridated) (%)	28.1	28.1	28.1	28.1	28.1
0.6 mg/L to 1.1 mg/L (fluoridated) (%)	71.9	71.9	71.9	71.9	71.9
**Sugary diet ^3^ (mean (SD))**	2.2 (1.9)	2.2 (1.7)	2.1 (1.6)	2.1 (1.5)	2.2 (2.0)
**Frequency of teeth brushing ^4^**					
Twice a day or more (%)	42.8	63.7 ^5^	63.7	64.5	62.7
Less than twice a day (%)	57.2	36.4	36.4	35.5	37.3
**Age started cleaning teeth ^1^**					
0–<7 months (%)	20.4	20.4	20.4	20.4	20.4
7–12 months (%)	57.1	57.1	57.1	57.1	57.1
13–24 months (%)	21.7	21.7	21.7	21.7	21.7
25–36 months (%)	0.7	0.7	0.7	0.7	0.7
**Dental service use in previous 12 months (%) ^6^**					
Caries in last 2 years, no service use			3.7	3.7	4.6
Caries in last 2 years, service use			9.4	21.0	26.1
No caries in last 2 years, no service use			60.5	38.6	32.3
No caries in last 2 years, service use			26.3	36.7	37.0

^1^ Time constant covariate that remains the same throughout all Waves. ^2^ Socio-economic position (SEP) was classified into quintiles reflecting relative SEP within the participating cohorts. ^3^ Higher scores indicate a higher frequency of reported consumption of sweet food and/or beverages in the previous 24 hours. ^4^ Due to rounding the data may not sum to 100. ^5^ Data inferred from Wave 4. ^6^ Dental service use not included in the regression model due to the 12 months recall period, data only available from Wave 4. Dental service use analysis in Figure 4.

**Table 2 ijerph-16-01970-t002:** Findings of multilevel mixed-effects logistic regression: Association with caries experience.

	Adj OR ^1^	95% CI		*p*
		Lower	Upper	
**SEP**				
Quintile 1 (high SEP)	1.00	-	-	-
Quintile 2	1.16	0.93	1.44	0.196
Quintile 3	1.34	1.06	1.69	0.013
Quintile 4	1.96	1.55	2.49	<0.001
Quintile 5 (low SEP)	1.92	1.49	2.46	<0.001
**Water fluoride level: Residential area in infancy**				
≤ 0.6 mg/L (non-fluoridated)	1.0	-	-	-
0.6 mg/L to 1.1 mg/L (fluoridated)	0.53	0.43	0.64	<0.001
**Remoteness**				
Major cities	1.0	-	-	-
Inner regional	1.09	0.90	1.32	0.393
Outer regional	1.33	1.06	1.66	0.013
Remote	0.82	0.44	1.52	0.529
Very remote	2.41	1.02	5.78	0.047
**Frequency of teeth brushing**				
Twice a day or more	1.0			
Less than twice a day	0.88	0.76	1.02	0.088
**Age started cleaning teeth**				
0–<7 months	1.0	-	-	-
7–12 months	0.94	0.77	1.17	0.595
13–24 months	1.01	0.77	1.32	0.958
25–36 months	1.21	0.38	3.89	0.749
**Sugary diet**	1.02	0.97	1.06	0.459
**Age in months**	1.02	1.02	1.03	<0.001

^1^ Odds ratios are adjusted for all explanatory factors in the model.

**Table 3 ijerph-16-01970-t003:** Findings of multilevel mixed-effects logistic regression: Association with dental caries including SEP and fluoride interaction.

			Adj OR ^1^	95% CI		*p*
				Lower	Upper	
**SEP**	**+**	**Water fluoride level**				
Quintile 1 (high SEP)	+	Fluoridated	1.00	-	-	-
Quintile 2	+	Fluoridated	1.00	0.78	1.30	0.975
Quintile 3	+	Fluoridated	1.24	0.94	1.65	0.129
Quintile 4	+	Fluoridated	1.93	1.45	2.55	<0.001
Quintile 5 (low SEP)	+	Fluoridated	1.54	1.14	2.07	0.005
Quintile 1 (high SEP)	+	Non-fluoridated	1.22	0.87	1.72	0.251
Quintile 2	+	Non-fluoridated	2.25	1.56	3.25	<0.001
Quintile 3	+	Non-fluoridated	2.27	1.64	3.14	<0.001
Quintile 4	+	Non-fluoridated	3.04	2.15	4.31	<0.001
Quintile 5 (low SEP)	+	Non-fluoridated	4.06	2.88	5.74	<0.001
**Remoteness**						
Major cities			1.0	-	-	-
Inner regional			1.09	0.89	1.32	0.399
Outer regional			1.33	1.06	1.65	0.013
Remote			0.82	0.45	1.50	0.516
Very remote			2.33	1.00	5.42	0.049
**Frequency of teeth brushing**						
Twice a day or more			1.0	-	-	-
Less than twice a day			0.88	0.76	1.02	0.085
**Age started cleaning teeth**						
0–<7 months			1.0	-	-	-
7–12 months			0.94	0.76	1.16	0.580
13–24 months			1.00	0.77	1.31	0.979
25–36 months			1.19	0.37	3.79	0.765
**Sugary diet**			1.01	0.97	1.07	0.413
**Age in months**			1.02	1.02	1.03	<0.001

^1^ Odds ratios are adjusted for all explanatory factors in the model.

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
