# Peer review of "The Impact of Policy Modifiable Factors on Inequalities in Rates of Child Dental Caries in Australia"

_ijerph, 2019, doi:10.3390/ijerph16111970_

Round 1
Reviewer 1 Report
The paper does a fine job of documenting and analyzing the data from the national survey. The additional features including fluoridation, where the children reside, sugary diet, tooth brushing, access to services and age and gender are very significant and add power to the national survey, and consequently to the study. Your suggestion that the reliability of the national data should be further examined, is a major consideration. Also, I would not want further studies to be limited by the locations of the children, especially those in "very remote" areas. This could create a "domino" affect for upcoming studies to draw upon these same limits. Those children, who may or may not be indigenous, should be considered equally important. We can not assume that their oral health status would be the same as children from moderately remote locations especially when we look at diet in terms of access to sugary foods.
Would it be possible to explain more about how communities are fluoridated in Australia as some idea about the degree of naturally fluoridated resources?
Wondering why race and/or ethnicity were not included in your tables compared to what type environment the children and their families reside. In the discussion, you mentioned that the national data does not take into consideration whether the family or child had relocated. From a global and national perspective, I believe this is important to provide information, perhaps gathered from other studies that may add another interesting variable in terms of how geographic location, school systems, etc may also have an influence on the child's/family's oral behavior.
I would suggest that some of these items be incorporated in the discussion or conclusion.
Author Response
No. | Comment | Line number | Page number | Author response |
1 | The paper does a fine job of documenting and analyzing the data from the national survey. The additional features including fluoridation, where the children reside, sugary diet, tooth brushing, access to services and age and gender are very significant and add power to the national survey, and consequently to the study. Your suggestion that the reliability of the national data should be further examined, is a major consideration. Also, I would not want further studies to be limited by the locations of the children, especially those in "very remote" areas. This could create a "domino" affect for upcoming studies to draw upon these same limits. Those children, who may or may not be indigenous, should be considered equally important. We can not assume that their oral health status would be the same as children from moderately remote locations especially when we look at diet in terms of access to sugary foods. | 9 | It is an inherent limitation of LSAC that the sample is not representative of children in the most remote locations, nor of Aboriginal and Torres Strait Islander children. These limitations are inherent in the design largely for practical reasons.
We have noted this limitation in the discussion (lines 495-497, page 16):
Additionally, children from very remote areas and Aboriginal and Torres Strait Islander children are not well represented in LSAC and care should be taken in extrapolating from these findings. It is important that future studies should be as inclusive as is feasible and to ensure routine data systems extend to rural areas.
| |
2 | Would it be possible to explain more about how communities are fluoridated in Australia as some idea about the degree of naturally fluoridated resources? | 9 | We have added the following to lines 360-367, page 13:
During the waves of the cohort, over 70% of the Australian population received fluoridated water ranging from 0.6 to 1.1 mgF/L. Australia mostly has very low levels of naturally occurring fluoride. There are very few areas with natural fluoridation, mostly in remote or very remote areas (central Australia, remote areas of Queensland). Natural Fluoridation levels range from 0.5 to 1.7 mgF/L. In regions where levels of naturally occurring fluoride fall within Australia’s recommended range, water suppliers do not add fluoride to the water supply. (NHMRC review, 2017) The study sample did not include children from those areas.
| |
3 | Wondering why race and/or ethnicity were not included in your tables compared to what type environment the children and their families reside. In the discussion, you mentioned that the national data does not take into consideration whether the family or child had relocated. From a global and national perspective, I believe this is important to provide information, perhaps gathered from other studies that may add another interesting variable in terms of how geographic location, school systems, etc may also have an influence on the child's/family's oral behavior. | 9 | We do not have a lot of information about race and/or ethnicity in LSAC – only whether children are Aboriginal or Torres Strait Islanders and country of birth of parents and grandparents which doesn’t classify race and/or ethnicity. This is a limitation but is probably not especially relevant for this research question. This has been noted in the limitations in the discussion as explained in response #1.
In terms of universal provision of fluoridation this would not need to be tailored according to race or ethnicity – rather at a full population level by local area. |
Reviewer 2 Report
Line 39 “accumulation and retention of bacterial dental plaque” – I’m not sure that this is a problem in children
Lines 40-1 “factors affecting embryological formation of tooth germ formation” - this is not clear – rephrase
Lines 75-77 There is no reliable evidence that water fluoridation works to reduce health inequalities. The 1999 paper by Colwyn Jones and Helen Worthington was an ecological study unable to take account of the many confounding variables. Armfield’s study was cross sectional in nature, subject to similar limitations. The strongest evidence on the matter lies in the review. The review you refer to actually uses the Truman review and York review as its sources. The York review is often misinterpreted, so much so that the Centre that produced it issued a statement to clarify matters, one of these was “The evidence about reducing inequalities in dental health was of poor quality, contradictory and unreliable”. The Truman review refers back to the York review on the question of disparities. It is interesting that you have not referred to a more recent systematic review on water fluoridation – the Cochrane review from 2013. This review looks at effectiveness, which is clear, however, it is questionable as to whether this evidence is still applicable today considering that evidence of effectiveness comes from studies conducted at a time before the use of fluoride toothpaste was widespread. The review also reiterates that there is no reliable evidence that water fluoridation works to reduce health inequalities.
Reference 40 for the 2003 Cochrane fluoride varnish review – this review has been updated; please refer to the most recent review.
Lines 147-8 – there is a sentence with only one word.
Not sure that it is helpful to include the percentage of waves by quintiles in Table 1 since it is obvious that this will be 20%.
I think it is interesting that you did not find an association between reported brushing behaviour and caries. This could be due to problems with self-report data or possibly, I wonder if it would make a difference if you dichotomised the data differently – so as to compare regular brushing (once a day or more) vs non-regular brushing (less than once a day). The evidence around brushing is for regular brushing rather than for twice daily brushing (which is the recommendation). This is not something that I would like to see changed in your paper or even commented on but just a thought I had about this issue. Also critical if whether fluoride paste was used or not but I understand that this isn’t something that could be found in the data available.
Your finding that water fluoridation appears to ameliorate the social gradient in caries is very surprising indeed, this sits in contrast to all other multiple time point studies in which this question has been addressed. I would urge caution here, ensure the data is absolutely correct and has been dealt with appropriately. I am not a statistician but I am able to appreciate that you are dealing with an incredibly complex data set here. It does seem a little strange to me that the ORs in Table 3 for deprivation quintile and fluoride level for caries have all been compared to quintile 1 fluoridated. It would seem to make more sense to me to separate out the data but as I say I am not a statistician.
Section 4.2 needs a bit of work – at the moment it reads like a list of related studies but no real attempt is being made to critically tease apart the reasons why there are contradictory findings in the literature. I also find it confusing that a school based brushing programme seems to be being contrasted with a school screening programme – these are entirely different interventions with different aims and mechanisms of action.
Author Response
Please find our responses attached.

Round 2
Reviewer 2 Report
I have no further comments.